# Geometric Partition Entropy: Coarse-Graining a Continuous State Space

**DOI:** 10.3390/e24101432

**Published:** 2022-10-08

**Authors:** Christopher Tyler Diggans, Abd AlRahman R. AlMomani

**Affiliations:** 1Information Directorate, Air Force Research Laboratory, Rome, NY 13441, USA; 2Department of Mathematics, Embry-Riddle Aeronautical University, Prescott, AZ 86301, USA

**Keywords:** entropy estimator, information theory, ignorance, uncertainty

## Abstract

Entropy is re-examined as a quantification of ignorance in the predictability of a one dimensional continuous phenomenon. Although traditional estimators for entropy have been widely utilized in this context, we show that both the thermodynamic and Shannon’s theory of entropy are fundamentally discrete, and that the limiting process used to define differential entropy suffers from similar problems to those encountered in thermodynamics. In contrast, we consider a sampled data set to be observations of microstates (unmeasurable in thermodynamics and nonexistent in Shannon’s discrete theory), meaning, in this context, it is the macrostates of the underlying phenomenon that are unknown. To obtain a particular coarse-grained model we define macrostates using quantiles of the sample and define an ignorance density distribution based on the distances between quantiles. The geometric partition entropy is then just the Shannon entropy of this finite distribution. Our measure is more consistent and informative than histogram-binning, especially when applied to complex distributions and those with extreme outliers or under limited sampling. Its computational efficiency and avoidance of negative values can also make it preferable to geometric estimators such as k-nearest neighbors. We suggest applications that are unique to this estimator and illustrate its general utility through an application to time series in the approximation of an ergodic symbolic dynamics from limited observations.

## 1. Introduction

The concept of entropy has had far reaching impact across science since its inception over 150 years ago, first as a foundational concept in thermodynamics, leading to the development of statistical physics [1,2] and quantum mechanics [3], then later as an encoding limit and a more general quantification of uncertainty in information theory [4,5]. Entropy-based analysis tools have been applied in almost every field of science, e.g., economics [6], and the wide range of applications continues to grow today within data-driven science [7,8,9] and machine learning [10]. In spite of this overall success, a lingering issue has remained at the heart of the definition of entropy that hearkens back to a fundamental question of whether reality is discrete or continuous.

The geometric partition entropy is an estimate of entropy as understood through the more recent interpretation as a measure of ignorance [11,12,13]. In the context of continuous state spaces, this change in perspective can be summarized in the terminology of thermodynamics by thinking of a data set as actual observed “microstates” of the system (assuming total accuracy in measurement); entropy-based analysis is then somewhat of a reversal of classical entropic analysis in that a set of “macrostates” is sought to represent the underlying phenomenon as a coarse-grain model, and the concept of entropy then quantifies the uncertainty in using this model to predict the unknown phenomenon.

Over the past century and a half, the topic of entropy has had a reputation for being akin to quicksand, in that the harder one seeks solid ground, one often suspects they are losing their footing. Therefore, while we attempt to provide a new foundation for the study of entropy on continuous state spaces, we must admit it is likely impossible to provide the full picture of this storied history. Since we are unable to resolve the fundamental question of whether reality is discrete or continuous at the smallest of scales (say, on the planck length), in this article, we will assume that measurements of some macroscopic phenomenon can be considered truly continuous. We will rely on past arguments that both the thermodynamic understanding of entropy and the information-based adaptations are theoretically only applicable to discrete state spaces [14,15]; and regardless of the utility of commonly used generalizations as approximations, these concerns justify seeking a more effective theory for continuous states spaces.

In Section 2, we begin by providing a summary of the origins of entropy and the arguments for its fundamentally discrete nature, which supports a transition away from traditional entropy measures in the case of continuous state spaces. We then provide context in Section 3 for where our new entropy estimate fits within the literature, before defining its simplest method of computation in Section 4. Through a range of samples from example data distributions in Section 5, we show that the geometric partition entropy outperforms the basic histogram-binning method in terms of consistency and informativeness and avoids the potential for spurious negative results that can plague other geometric estimators such as k-nearest neighbors (knn). Although it may not always be preferable to knn, we show in Section 6 that the computational complexity of our estimator indicates utility for large data sets. It is the basic one-dimensional version of knn presented in [16] that we focus on here, but future work will seek to develop a higher dimensional version of geometric partition entropy to enable a more full comparison with the many celebrated knn-based estimators used in the context of estimating mutual information. We go on in Section 7 to outline a few unique applications for this approach to entropy and show the general power of this method for data-driven science through an application in time series analysis, where the entropy of a long chaotic time series can be efficiently estimated from small samples and a more ergodic Markov chain approximation can be generated from a limited sample of observations from a dynamical system on a continuous state space.

## 2. The Discrete Nature of Existing Theories of Entropy

### 2.1. Boltzmann and the Density of States

While the atomistic viewpoint eventually provided theoretical support and predictive power to Clausius’s continuum-based theory of thermodynamics, it was initially met with rejection by the scientific community [13,17]. The main opposition centered on the fact that the second law of thermodynamics broke the time symmetry of Newton’s laws, while the atomists sought to use those very same laws to explain the arrow of time’s origin, i.e., the reversibility paradox. In the face of that criticism, Boltzmann and others rightly identified that measurable quantities of an apparently continuous medium could be explained by statistically averaging over an associated property of an enormous number of constituent atoms. For example, the temperature of a confined gas can be identified with a multitude of atomic/molecular kinetic energies, and because the number of atoms/molecules is considered to be on the order of Avogadro’s number, the statistical treatment results in distributions of the underlying properties that are sharply peaked around typical values, leading to well-defined and predictable outcomes. However, conflict arose from the averaging process itself, as each possible kinetic energy (i.e., a macrostate) can be associated with a combinatorial number of potential microstates, which are fundamentally unmeasurable. Incorporating this “degree of broadening” of the macrostates (as it was called in [18]) is vital to obtaining the correct average, and Boltzmann’s definition of entropy assumes a particular distribution of these microstates, namely that they are all equally likely.

Recent developments in the study of negative temperatures seem to confirm that the Boltzmann definition [19,20] of entropy is more fundamental than Gibbs’ [21,22], and it is Boltzmann’s formulation that Max Planck showed holds only for discrete energy states [2]. Thus, it may be that the common practice of computing thermodynamic entropy using an integral approximation, which relies on a functional density of states to represent the statistical weighting, may have led to a somewhat laissez faire attitude in applying the Gibbs definition more generally to continuous state spaces in the following years.

For simple systems such as a container of an ideal gas, where the macrostates for energy are countable and finely delineated, such a density of states can be defined to approximate the degree of broadening, e.g., the limiting density of the integer valued points in *k*-space (i.e., wave number space), and enable accurate integral estimates of thermodynamic quantities; but, we wish to impress upon the reader that this successful integral approximation is due to the density of states roughly associating the correct discrete combinatorial number of microstates with each of the very large (or countably infinite) number of energy macrostates, meaning the success is likely due to the vanishing distances between the macrostates rather than a true limiting underlying distribution over a continuous state space.

Furthermore, from the classical statistics view for thermodynamic entropy, it has been stated that, “the concept of the ‘number of microscopic states’ cannot be defined at all [18]”, and also that entropy is only defined to within an additive constant (based on the chosen units), meaning differences in entropy over a process are the only useful quantities [18,23].

### 2.2. Shannon and the Principle of Insufficient Reason

In contrast, the fundamentally symbolic nature of digital communications enabled the analysis of a discrete set of macrostates that were free from any underlying hidden phenomena (microstates). Thus, in keeping with Laplace’s Principle of Insufficient Reason, the assumption of a uniform density of states over the discrete set of macrostates was the obvious choice [24]. Shannon then proved that for a probability distribution *p* on a discrete set of *M* states, the quantity that is positive, increases with increasing uncertainty, and is additive for independent sources of uncertainty, is in fact
(1)$$H\left(p\right)=-K\sum_{i=1}^{M}p_{i}\log\left(p_{i}\right),$$
which mirrored Gibbs’ discrete definition of entropy as $-k_{B}\langle \log\left(p\right)\rangle$, hinting at the potential unification of the two theories under the interpretation of information. The maximum-entropy principle from [5] was provided as a justification for the assumption of ergodicity in the absence of microstates, and the use of the constant K=1/log(2) was chosen to essentially use a base two logarithm aligning information theory with the rise of binary-based computers.

While Shannon’s generalization of his discrete theory into integral form has since been widely applied to continuous state spaces, the abstraction tacitly retained the assumption of ergodicity in the limiting equally sized, but infinitesimal, macrostates. In regard to this detail, Shannon provided the less than convincing argument that “[t]he occasional liberties taken with limiting processes in the present analysis can be justified in all cases of practical interest [4]”. This basic assumption led to the histogram-based binning approximation using Riemann sums, which ignores the potential for a density of states multiplier. Similar to thermodynamics, Shannon also pointed out that “[t]here is one important difference between the continuous and discrete entropies. In the discrete case the entropy measures in an absolute way the randomness of the chance variable. In the continuous case the measurement is relative to the coordinate system [4]”.

Finally, it was pointed out in [5], that the coincidental use of the sum (Equation 1) in both thermodynamics and information theory need not imply a direct relationship between the two concepts. In fact, we believe that the complexity of physical reality implies that information theory will only ever approximately align with thermodynamics. Regardless, the fortuitous assumption of quantized energy levels, which enabled Max Planck’s success by removing the problematic infinities for the determination of probabilities [2], and the discrete symbolic representation of digital communications leaves both theories limited to the context of discrete state spaces. It is admitted as a footnote in [5] that “the problems associated with the continuous case are fundamentally more complicated than those encountered with discrete random variables”.

The information theory definition of entropy is more general, but as the application of entropy-based methods to data science and machine learning has grown, we have perhaps forgotten that the Principle of Insufficient Reason does not apply to data sampled from a continuous observation space. We define the geometric partition entropy as a measure of ignorance [11,12,13] in prediction of large scale outcomes, given a limited sample of observations. In that context, all existing parametric and non-parametric entropy estimation techniques in the literature will suffer from undue influence by the once reasonable maximum entropy principle. Interestingly, in contrast, the geometric partition itself provides a coarse grain representation of the phenomenon as a set of equally probable macrostates, i.e., a maximum entropy set.

## 3. Traditional Entropy Estimators

Within the vast literature devoted to estimating entropy, there are three main categories of traditional approaches to estimating the Shannon entropy of an underlying phenomenon from which a number of sample observations have been measured; and each has its limits: (i) histogram-based binning methods, which require a large number of observations for consistent results and has the important question of how to choose the bins; (ii) geometric methods, such as *k*-nearest neighbors (knn), which are more robust for limited sample sizes, but are computationally intensive and can even result in nonsensical negative values; and finally, (iii) parametric methods, which are more aligned with the treatment of continuous microstates and essentially assume a density of states based on a priori knowledge of the underlying distribution from which the data were sampled.

While histogram estimates are known to be less accurate in higher dimensions, they are still effective and are used widely in the one-dimensional setting, where Equation (Equation 1) is utilized on the set of proportions from the normalized histogram counts. For simple distributions, our new estimate agrees to a large extent with histogram-based methods. However, our method provides various advantages when applied to samples from complex distributions and those with outliers when compared to either histogram-binning or geometric estimators as we will show.

Although there are many knn-based estimators that have grown out of the celebrated KSG estimator for mutual information [25], all of those techniques are only relevant to higher dimensional data and are quite different in their implementation from the one-dimensional version that was introduced in [16] as a generalization of the original nearest neighbor estimator by Kozachenko and Leonenko in [26]. The one-dimensional knn estimator is computed for a sample $X_{N}={\left\{x_{i}\right\}}_{i=1}^{N}$ as
(2)$$H_{k}^{NN}\left(X_{N}\right)=\frac{1}{N}\sum_{i=1}^{N}\log\left(N\rho_{i}^{k}\right)+\log\left(2\right)-\psi \left(k\right),$$
where ψ represents the digamma function and $\rho_{i}^{k}$ is the Euclidean distance from $x_{i}$ to its kth nearest neighbor [16]. Although it may appear at first glance that this estimator uses the distances directly, the lack of a minus sign in the equation indicates that the argument of the logarithm can be considered to be the reciprocal of the distances used, meaning sparsely populated regions will correspond to smaller arguments in the logarithm.

While the third category seeks to find the expectation value of the logarithm of a distribution and does indeed provide a well-defined value, they are only useful for quantifying changes in entropy due to the potential for the changing of variables mentioned in Section 2.1.

There is a fourth, less widely known category of entropy estimation techniques based on spacings [26,27,28,29,30,31,32] and so-called quantile-based “plug in” estimators [33]. The quantile-based estimators seek to approximate the differential entropy using a quantile density function [34,35] and so these are not as relevant to our discussion, but the spacing-based estimators are similar in spirit to the geometric partition entropy in that the lengths between data points is central. The most popular of these is the *m*-spacings approach [28,29,30,31], but it is actually quite similar to the one dimensional version of knn and was shown to be inferior to knn in [16], where it was also shown that knn is usually optimal under the choice of k=4.

The method of *m*-spacings also uses the reciprocal of the lengths in question, meaning that although both use distances between points, they still seek to approximate the entropy as the expectation of a probability density, whereas our density measures the ignorance associated with the spaces more directly, which sets our estimator apart from what has come before.

It may not be surprising that the uncertainty about one’s ignorance is equivalent in some sense to the uncertainty about one’s knowledge of a data set, and entropy as a measure of ignorance has become a common interpretation in information theory [11,12,13]. Despite this difference in viewpoint from traditional estimates, due to the nature of spacings for well-behaved data distributions [27], we will see that our estimator often agrees with traditional approaches, but offers certain advantages for more complex data sets.

We should also point out that the knn (and *m*-spacings) estimator is what may be called a local estimator, in that it uses *N* (or N−2m) distinct, but essentially overlapping macrostates to produce the estimate. This means that no direct association can be made between the number of neighbors *k* (or *m*) and the parameters that are used to define either histograms bins or quantiles. As such, we will focus our comparisons between geometric partition entropy and histogram-binning, but will include the value obtained by knn for k=4 in each plot for reference.

## 4. The Geometric Quantile Partition Entropy

We use the term geometric quantile partition entropy to describe the approach defined in this section, but one can define the partition itself in many alternate ways; meaning the more generic term geometric partition entropy may be applied to adaptations of this approach, wherever the partitioning process incorporates the geometry of the sample into its computation. However, we must emphasize that this is not to be confused with the geometric entropy that has been used in quantum mechanics [36].

Although the concept of quantiles effectively captures the essence of the idea, in the spirit of Riemann sums, more intricate partitioning procedures have been explored for providing marginally better results (analogous to say Simpson’s rule versus midpoint rule for integration); however, we will focus on the basic quantile-based definition for simplicity.

Assuming *N* observations in R, e.g., ${\left\{x_{i}\right\}}_{i=1}^{N}$, and given a particular choice of *k*, we define a partition of the data set by computing the i/k-th quantiles of the data set for i=0,1,...,k, denoting them as ${\left\{q_{i}\right\}}_{i=0}^{k}$, where the minimum and maximum quantiles (denoted by $q_{0}$ and $q_{k}$) are taken to be $x_{1}$ and $x_{N}$ respectively, in the absence of any additional information. We call this set of quantiles (including the end points) a *quantile partition* of the data set, and we define a probability distribution using the geometry of the partition to represent the ignorance associated with using the coarsely defined discrete states. Letting the overall range of observations be denoted by $L=q_{k}-q_{0}=x_{N}-x_{1}$ and taking the interval lengths between consecutive quantiles to be $l_{i}=q_{i}-q_{i-1}$ for i=1,...,k, we define the ignorance probability distribution *p* by the proportion of these lengths,
(3)$$p={\left\{\frac{l_{i}}{L}\right\}}_{i=1}^{k}.$$

Since each interval of the partition will contain roughly the same number of data points, those with larger lengths, $l_{i}$, necessarily have a higher “density of ignorance” with respect to the phenomenon. The *geometric partition entropy* then quantifies the distribution of ignorance by the sum (Equation 1) using this ignorance probability distribution. We use K=1/log(2) for direct comparison with Shannon entropy and so can compute (Equation 1) as
(4)$$\begin{array}{cc} H_{k}^{GP}\left(p\right) & =-\sum_{i=1}^{k}p_{i}\log_{2}\left(p_{i}\right)\\ \; & =-\sum_{i=1}^{k}\frac{l_{i}}{L}\log_{2}\left(\frac{l_{i}}{L}\right)\\ \; & =\log_{2}\left(L\right)-\frac{1}{L}\sum_{i=1}^{k}l_{i}\log_{2}\left(l_{i}\right). \end{array}$$

Interestingly, using the Boltzmann constant, $K=k_{B}$, would simply give the one dimensional version of Boltzmann’s definition of entropy in terms of phase space volumes [19] (where we have used the quantile partition to define unequal volumes from the observation space). Similar arguments to those provided by Planck in the context of thermodynamics for the requirement of finite probabilities will then hold here as well; since the constant 1/L can be factored out of the summation, the overall measure of the observation space, *L*, must be bounded in some way. This implies the same potential for constant multipliers and supports the concept that differences in any entropy estimate are the useful quantities. We have chosen to define *L* by the span of our observations for simplicity, but in certain applications, there might be well defined bounds and/or methods of determining more reasonable values for $q_{0}$ and $q_{k}$ other than the end points.

We also note that if there are repeated values in the data set, some of the quantiles might be equivalent, meaning the discrete distribution may have some entries with zero “probability” (i.e., ignorance), effectively leading to less than *k* “macrostates”. Furthermore, the most common methods for computing the quantiles utilizes interpolation, meaning there may also be macrostates whose associated lengths are nonzero (indicating nonzero probability), even though no sample data are represented. Recall that this approach measures the uncertainty in the unsampled space: when values are repeated, there is essentially no uncertainty; while if the number of quantiles sought is larger than the number of *unique* values, there may be reason to split up larger spaces into multiple states; this point is precisely where more intricate partitioning procedures might produce increased benefit in the future.

## 5. A Tale of Two Entropies: Geometric Partitioning vs. Histogram-Binning

The power of an entropy estimation technique should be sought in its consistency, and not necessarily in its agreement with other estimators, since as discussed in Section 2, it is usually differences in entropy that matter. Especially for continuous domains, we must reiterate that no true entropy value even exists for comparison due to the possibility of coordinate changes in differential entropy or unknown additive or multiplicative constants related to the definition of probabilities or the removal of biases. With this in mind, we will provide the knn estimate for k=4 for all figures in this section for reference, but we will discuss the benefits and drawbacks between our method and knn in Section 6.

Histogram-binning is still the most commonly used estimator for a wide range of disciplines, especially in the context of one-dimensional data. However, it is well known that in the limit of many bins, this estimator will converge to the uninformative value of log(N), and so the process of choosing the best number (and distribution) of bins is still an active area of research. A good choice is usually achieved through the definition of a loss function and repeated trials (or through some iterative Bayesian process), which leads to longer computational time, with still no guarantee of global optimality.

In the context of continuous state spaces, it is helpful to note the similarities between the histogram-binning procedure and that of geometric partition entropy. The first step in the process can be understood as defining a set of coarse grain macrostates. The classic histogram approach simply defines equally spaced bins among the observation range. The probability distribution used to compute the entropy is then the simple count of the data falling into each bin. There is a certain symmetry when compared to the idea of geometric partition entropy in that by defining the macrostates using the quantile partition, the traditional concept of bin counts would result in a maximum entropy distribution. Instead, we define a geometric distribution based on the sizes of the macrostates, and normalize this distribution by the size of the overall observation space.

We begin with two sets of sampled microstates from simple well-behaved distributions in order to illustrate the correspondence of geometric partition entropy with traditional histogram-based estimates. Figure 1 shows estimates for samples taken from: (a) a relatively small sample of N=200 points from a uniform distribution on the interval [0,1] and (b) a larger sample of N=5000 points drawn from a normal distribution with mean μ=0 and standard deviation σ=1. The histogram and geometric partition entropy estimates are plotted against the choice in number of bins or quantiles (k) for a direct comparison, while the knn estimate using k=4 nearest neighbors is included for reference.

The sample sizes have been varied to indicate the effects on the estimators, but in either case assuming a single macrostate leads to zero entropy as expected. Too few states in general will lead to a lack of representation for the true complexity of the distribution, and thus in general an underestimate of the entropy. Here, we see a general agreement between the two main metrics for simple distributions, but as the number of macrostates (*k*) increases, an indication of the improved consistency for geometric partition entropy over histogram-binning can be seen in the comparative smoothness of the lines in Figure 1a where the smaller sample size leads to instability with the choice of bins. The same trend holds for the normal distribution, but the larger range of values for *k* make this feature less apparent in Figure 1b. In terms of knn, we only point out that the estimate for Figure 1a is actually negative and therefore indicates one of the benefits of our geometric method over this estimator.

At the other extreme, as k→N, we see that the two estimates approach similar, but different values, which may indicate another benefit of our method. Using too many histogram bins (k>>N) for a data sample of unique values tends toward each bin having a single (or no) representative, and in fact, this extreme tends toward the uninformative maximum value of $\log_{2}\left(N\right)$ in all cases of histogram-binning. In contrast, the geometric partition entropy only converges toward $\log_{2}\left(N\right)$ for a truly uniform distribution; when defining a partition around each data point (k=N) or even using interpolation (k>N), the geometric partition entropy will always incorporate the sampling irregularities and will only converge as $\log_{2}\left(N\right)$ in the limit of infinite data sampled from a uniform distribution. More generally, the geometric partition entropy will approach some functional dependence on *N*, but that relationship will represent the distribution of empty space in the sample relative to the range of observations, which should result in a more informative limit, although again, we must remember the potential for constant multipliers from our assumptions on *L*.

Aside from the general agreement with traditional metrics, the advantages of the geometric partition entropy over histogram estimates become more apparent as we consider more complex distributions, smaller sample sizes, and samples having extreme outliers, where the idea of quantifying the ignorance over the unsampled region is helpful.

While Figure 2a shows a similar correspondence with histogram-binning for multi-modal distributions such as the overlapping mixture of three Gaussians; Figure 2b shows that for smaller samples from the same distribution, the histogram-binning estimate oscillates about a general trend line, while the geometric partition entropy is more consistent with variation of *k*, meaning the results are not as sensitive to the choice in *k*. Figure 2c shows that even a few extreme outliers (in this case perturbing five points from a sample of N=5000 points drawn from a normal distribution by multiplying them by 100) leads to erratic results for smaller numbers of bins in histograms, while the geometric partition entropy estimate not only remains smooth, but converges rather quickly toward a stable estimate (which coincidentally qualitatively agrees with the knn estimate) as the outliers are the feature indicating the overall level of ignorance. Furthermore, the inset shows that the geometric partition entropy estimate is more robust against a growing value for *k* in the presence of outliers, since histogram-binning will generally involve many bins with no representatives, which in turn will not be involved in the computation of entropy.

Finally, Figure 2d shows that the geometric partition entropy can perform better (in terms of qualitative agreement with knn, which is consistent with expectations of scale-free distributions) as k→N, even under a small sample size, e.g., N=50 points, drawn from a fat-tailed distribution such as the Pareto distribution. One may be concerned by the fact that the geometric partition entropy estimate has a larger range of values and is eventually much larger than histogram-binning, but as this is an example of a scale-free distribution, the level of ignorance should be large, and in fact unquantifiable to an extent due to a lack of mean in the distribution. The inset shows that one would have to use on the order of k= 20,000 for the histogram-based method to begin to capture similar levels of entropy since even for extreme values of *k*, most of the data will fall into the first bin. In contrast, the geometric partition entropy provides an indication of the complexity that arises from such distributions since even for k=50 the entropy estimate includes information from every observation, though limited by the sample for obvious reasons.

## 6. Geometric Accuracy without Increased Computational Complexity

While the high dimensional versions of knn have recently garnered a lot of attention for providing consistent entropy estimates for the computation of mutual information between long time series, they are well known to have drawbacks including the curse of dimensionality as the dimensions increase and the problem of bias that can lead to negative values as in the one dimensional case.

The geometric partition entropy in its current form is only applicable to state spaces of one-dimension, and so the number of knn estimators that are appropriate for use in this context is limited. Due to the very different methods of computation, the entropy estimates that result from knn tend to be quite different from either our method or histogram-binning. This is often due to the issues that were raised in the first few sections of this article, namely the potential for additive and multiplicative constants. However, the fundamentally different approaches used also means that direct comparisons based on the parameter values has limited utility. In the absence of any ground truth for comparison of accuracy, we must resort to consideration of consistency and qualitative features.

The knn estimator is known to be very consistent, and in fact, often does not change much for a large range of *k* values (meaning k=4 is generally representative of other choices). It is also known for being useful on small samples. Despite these obvious strengths, the geometric partition entropy still has two possible advantages over such estimators: (i) its computational simplicity and (ii) its lack of bias, meaning it is never negative.

In the context of one dimensional continuous data, *k* quantiles can be obtained in O(k·N) time, and even if sorting is used in this process (as in MATLAB’s quantile function), on average it will be O(Nlog(N)) time complexity, which is an improvement upon the quadratic complexity of knn and other local averaging estimators.

More work and further analysis will be required to quantify the benefits of geometric partition entropy over other geometric approaches, especially in the context of estimating mutual information for high dimensional data, but we believe that the basic idea behind geometric partition entropy will enable the use of the geometry without the excessive computational burden found in these often more intensive estimators.

## 7. Application: Time Series Analysis

The use of geometric partition entropy has shown significant improvements in multiple applications. We will illustrate this through a relatively simple task in time series analysis, but this method is likely to provide similar benefits to any application of entropy in continuous state spaces, particularly in sparse data environments due to its utilization of the sample geometry and increased sensitivity to outliers (see Figure 2). For example, we have identified an application in spectral clustering, namely automating the identification of a spectral gap, which is a common problem in this unsupervised machine learning technique. Our approach offers an entirely unique opportunity in this respect due to its validity for extremely small samples, e.g., two data points at a time, which may be indicative of a broader appeal for this approach. That application was in fact the origin of this work, but it relies on a more intricate partitioning process and so we leave that for future publications to avoid unnecessary confusion.

Additionally, the normalized ignorance density distribution that is the main innovation in this approach may be used in place of the usual probability density in the definition of information theory measures such as the Kullbeck-Liebler divergence and mutual information, which is likely to lead to improved performance of network and causation inference algorithms. Future work will compare such results directly with the broader class of high dimensional knn-based methods in the context for which those estimators were designed.

However, for simplicity of presentation, we have chosen an application in time series analysis that illustrates the general utility of this approach with only a minimal introduction of new concepts. When analyzing dynamical systems from time series data, it is common to seek a discrete set of states in order to define a symbolic dynamics that can approximate the true unknown dynamics. If given a number of discrete states that span the set of possible observations, a state transition matrix can be approximated using the observed transitions from a sample of the time series, which effectively defines a Markov chain approximation to the dynamics. The critical step in this discretization process of a continuous state space is the definition of the boundaries of the coarse-grained macrostates. We will show that using the quantile-based partition is superior to other choices, and defines a set of macrostates that approximate a maximal entropy set.

We consider $N=10^{7}$ iterations of the logistic map and compute the empirical entropy of the time series under the choices of k=25 and k=500 states using the geometric quantile partition and histogram-binning estimators to show the variation in coarse graining. In order to determine the number of iterations of the time series needed to accurately estimate the overall entropy values of the larger data set, we consider the entropy estimates obtained from sub-samples of increasing size (scaled by the number of states, i.e., $N_{i}/k$ with $N_{i}\leq N$). Figure 3 shows that the quantile-partition entropy provides a better estimate of the overall entropy with very few data points: in (a) for a small number of states, the histogram requires approximately double the sample size to achieve the same level of utility that (due to interpolation) the quantile partition approach can estimate using a sample less than the number of states; and in (b) we see that for more dense state spaces, the histogram approach only stabilizes after using a sample whose size is approximately four times the number of states.

Perhaps more importantly, for a specified number of states, the quantile partition defines a more ergodic dynamics as compared to the equal size states associated with basic histograms. The resulting transition matrix is then more informative for the level of discretization. For reproducibility, we consider the first N=100 iterations of the logistic map using the initial condition of $x_{0}=0.1$ in Figure 4, where (a) shows the data set together with the histogram bins and quantile partition that are associated with the choice of k=25 macrostates. Using this small sample of the dynamics, we can then create state transition graphs by recording which state transitions have been observed. These directed graphs were generated using an adaptation of *circularGraph* [37] and are provided in Figure 4b,c for the histogram-binning and geometric partition respectively, where the node color indicates the origin of the edge and isolated nodes are highlighted red. The accompanying transition matrices, which include the observed proportions of transitions, are shown in (d) and (e) respectively, where the high probability transitions, the isolated nodes, and numbers of possible transitions all indicate problems associated with using equally spaced states.

Although the particular choice of initial condition that produced the state transition graphs in Figure 4 was intended to illustrate the sometimes extreme differences when defining coarse grain macrostates, these results are not atypical. Figure 5a shows that, even when averaged over time series from 1000 different initial conditions, the minimum degree for the histogram-based graphs are often less than unity, meaning there will often be at least one isolated node (i.e., unvisited state). This trend persists until the number of iterations used reaches approximately $N_{i}/k=10$; this is in contrast to only $N_{i}/k=2$ required for the geometric partition to generally avoid isolated nodes. Furthermore, the fact that the minimum degree converges to one for histogram style binning indicates that there will almost always be some of the equal size macrostates that have completely predictable (and thus uninformative) transitions. When considering the maximum degree of the same transition matrices in Figure 5b, we see that while the maximum degree for equal size macrostates will continue to increase as the sample size of the data increases, the quantile partition reaches a maximum early on at around $N_{i}/k=5$ and then as more data are included, the average maximum degree actually reduces, meaning the macrostate transitions tend to have higher local entropy.

## 8. Conclusions

A new estimator for entropy has been provided in the context of a one dimensional continuous state space. It has been shown to outperform the traditional histogram-binning approach in consistency and its ability to handle outliers and small samples. Although more work must be carried out to provide a comprehensive comparison with other geometric estimators in the high dimensional setting where they have been so useful, the computational simplicity of this new method provides a potential advantage over even these well known and celebrated approaches. The geometric partition entropy analysis seeks to inform a more data-driven way of defining a coarse-grained model of a continuous phenomenon and further to estimate the level of ignorance about the underlying phenomenon by using that coarse-grained model.

The quantification of uncertainty is achieved by concentrating a probability distribution on the regions of the measurement space where fewer measurements have been obtained, meaning less knowledge of the observable is available, and we call this probability distribution an ignorance density distribution. In addition, utilizing the uncertainty of the unknown will lead to new promising algorithms for change detection and machine learning as indicated by our initial application in spectral clustering. We have provided an important, but straight forward application that illustrates both the utility of this approach to defining a set of representative macrostates and showcases the power of the geometric partition approach for data-driven science under the restriction of small sample sizes.

## Figures and Tables

**Figure 1 entropy-24-01432-f001:**
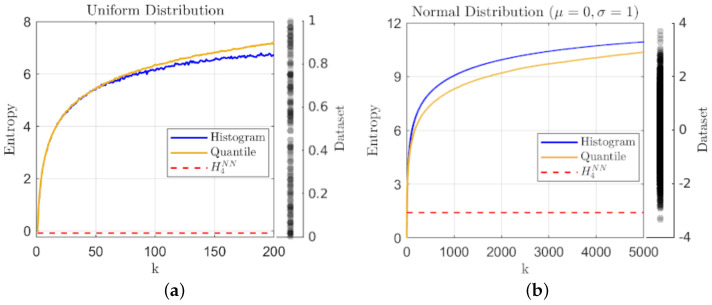
A plot of entropy estimates using the geometric quantile partition (**—**) and histogram-binning (**—**) methods as a function of the number of quantiles and bins respectively, together with the knn estimate (**- - -**) using k=4 for: (**a**) a sample of N=200 observations from the uniform distribution on [0,1] and (**b**) a sample of N=5000 observations from the normal distribution with μ=0 and σ=1; the data samples are included to the right of each plot for reference.

**Figure 2 entropy-24-01432-f002:**
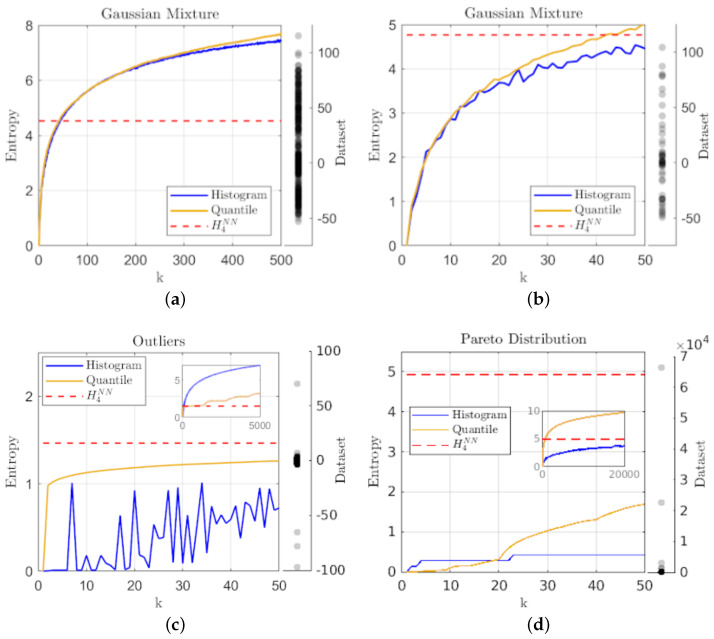
A plot of entropy estimates using the geometric quantile partition (**—**) and histogram-binning (**—**) methods as a function of the number of quantiles and bins (*k*), together with the knn estimate (**- - -**) using k=4 for: (**a**) a sample of N=500 points drawn from an overlapping mixture of three Gaussians; (**b**) a smaller sample of N=50 points drawn from the same overlapping Gaussian mixture; (**c**) a sample of N=5000 points drawn from a single Gaussian distribution with five extreme outliers manually introduced; and (**d**) a small sample of N=50 points drawn from a Pareto distribution to show the effectiveness on small samples from fat-tailed distributions.

**Figure 3 entropy-24-01432-f003:**
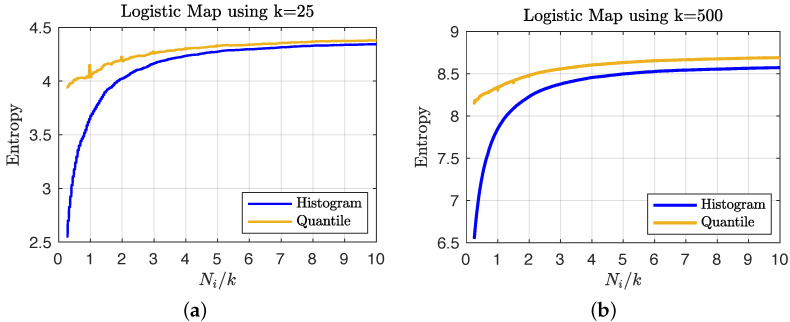
A plot of entropy estimates (averaged over 100 random sub-samples) using the geometric quantile partition (**—**) and histogram-binning (**—**) methods as a function of the ratio of sample size to the order of the state space (i.e., $N_{i}/k$) for the choices of (**a**) k=25 and (**b**) k=500 coarse grain macrostates.

**Figure 4 entropy-24-01432-f004:**
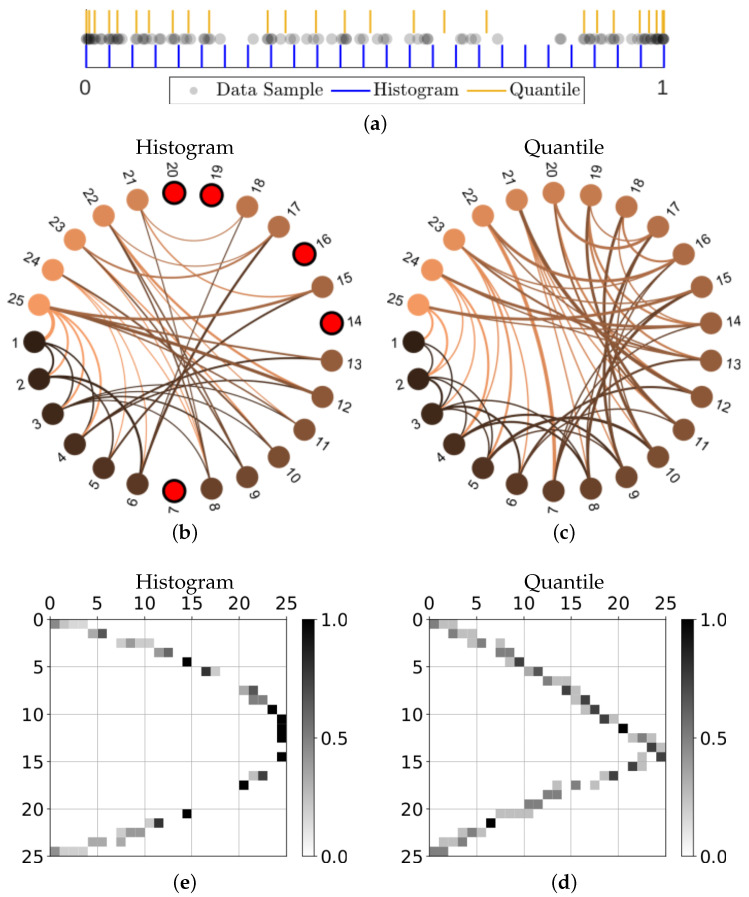
Taking N=100 iterations of the logistic map using the initial condition of $x_{0}=0.1$, we show: (**a**) the data together with the partitions defined by the equal sized macrostates (histogram-binning) and the geometric quantile partition; the state transition graphs approximated by the observed transitions using either (**b**) the equal size macrostates or (**c**) the quantile partition macrostates, where directed edges are colored based on the origin node and isolated nodes are highlighted red; and the corresponding matrices of transition probabilities in (**d**,**e**) respectively. Graph representations in (**b**,**c**) were created using *circularGraph*, Copyright (c) 2016, The MathWorks, Inc., adapted with permission from Ref. [37].

**Figure 5 entropy-24-01432-f005:**
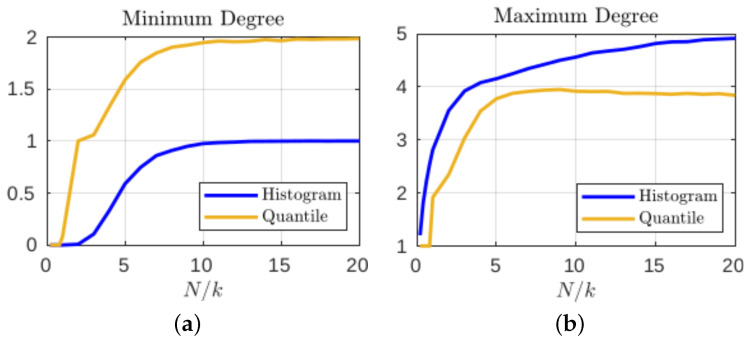
The (**a**) minimum and (**b**) maximum degrees in the resulting state transition matrices (averaged over 1000 samples of $N_{i}/k=4$ iterations).

## Data Availability

Data and accompanying implementations in MATLAB are available at https://github.com/tylerdiggans/GeometricPartitionEntropy.git, accessed on 12 August 2022.

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
