# Peer review of "Geometric Partition Entropy: Coarse-Graining a Continuous State Space"

_entropy, 2022, doi:10.3390/e24101432_

Round 1

Reviewer 1 Report

Report on the paper titled "Geometric Partition Entropy: Coarse-graining a Continuous State Space" submitted to Entropy

Summary: In this paper, a new entropy estimator has been presented in the context of a continuous one-dimensional state space. It has been demonstrated to perform better than the conventional histogram binning method in terms of consistency, handling outliers, and small sample sizes. Although probably not as effective as other geometric estimators (such as knn), this novel methodology has advantages over even existing well-known and well-recognized methods due to its computational simplicity. In order to define a coarse-grained model of a continuous phenomenon more data-drivenly and to evaluate the degree of ignorance about the underlying phenomenon using that coarse-grained model, geometric partition entropy analysis was developed.

Evaluation: The paper is very well written and comprehensive for a wide range of audiences. However, it is not very original (there are no mathematical developments) and the novelty is very limited. There is only a lot of text and direct applications. I am not sure that it is enough for a top-ranked journal such as Entropy. This can be a short conference paper, but I am skeptical about considering it as a research paper. 

Reviewer 2 Report

·         Re-Write the heading “ 2. Discrete vs Continuous”, in comprehensive manner.

·      In equation 1 where from  start., and define N.

·         Write down the formulas of Traditional Entropy Estimators in section 3

·         Some time I varies 1 to k and some time from N., clarified it.

·         Why chose Unifrom, normal and Preto distribution, explain the reasons.

·         Add n details, about the actual practical application in several fields  in section 7

Reviewer 3 Report

Major remarks:
1. In your paper, you mainly compare the proposed approach with a binning estimator for entropy. However, your approach is the geometric approach which should be compared with knn. For example, in my opinion, you should add this estimator to Fig 1 and Fig 2 and discuss the results. The same applies to the example in Section 7.
2. Implementation. The link provided does not work, and the Authors' repository for this paper is essentially empty.

Minor remarks:
1. Fig 1: Why do you have different sample sizes for uniform and normal distribution?
2. Fig 2: I would like to see more information about mixture distribution. Also, you should show how did you introduce outliers to Gaussian distribution.

Round 2

Reviewer 1 Report

I am happily surprised by this revised version, which is so much more exciting to read and makes clear the contribution of the findings. I have no new comment. My new decision is accepted.

Reviewer 2 Report

The author(s) successfully Revised.  

Reviewer 3 Report

The authors responded sensibly to all my objections. Thank you and I have no further comments.